# State-Space Modeling of Housing Sentiment for Regressing Changes of Real Estate Prices Following Short-Term Control Policy in China

**Taiyi Zang** [1] and **Hongmei Gu** [1,2,*]

1   School of Economic, Jilin University, Changchun 130015, China; cangty19@mails.jlu.edu.cn
2   Center for Chinese Public Sector Economic Research, Jilin University, Changchun 130015, China
*   Correspondence: nina_jlu@126.com

**Abstract:** Government may need to launch policies to stabilize real estate prices being away from unusual rise at an unexpected pace through short-term regulations of sales and purchases. Short-term control policies are often not effective immediately after withdrawal, but their effect easily attracts swift and intensive responses of consumer sentiments. The change in sentiment synchronizes with that of expectations, which together account for housing price in response to restrictions following short-term policies. The research objective of this study is to establish the role of housing sentiment in policymaking to regulate and stabilize real estate prices. To cope with the tough tissue of unclear knowledge about customers' sentiments, we employed the state-space model to explore the impact of short-term regulatory policies on housing sentiment. The research objective of this study also involves optimizing the instrument for assessing housing sentiments. Results showed that: Firstly, the short-term regulation and control policy enhanced positive sentiment in the housing market. Secondly, high positive sentiment further increased the cyclical prices. Thirdly, the upsurge of consumer sentiment has weakened the impact of short-term control policies on real estate market price. Lowered housing sentiment resulted in a reduction in the effectiveness of short-term control policies. Overall, our study verifies that high positive consumer sentiments will result in an increase in housing prices, hence it is customers' sentiments that caused the failure of short-term control policies.

**Keywords:** real estate short-term regulation; housing sentiment; cyclical real estate prices; state-space model; Markov switching variation model; price adjustment

## 1. Introduction

Since 2010, the Chinese government has introduced a series of short-term policies to enforce restrictions on purchases and sales to curb the uncontrolled rise in housing prices. It is still controversial among academic arguments whether the short-term regulation of policy intervention can stabilize real estate prices. Since the 2010s, short-term real estate control measures have succeeded in limiting the rise in housing prices [1–5], which was attributed to the elasticity of the supply-and-demand relationship. From the perspective of elasticities in supply and demand, short-term regulatory policies were found to have a significant impact on the relationship between supply and demand [6–8]. In contrast, at least parts of short-term regulations rarely limit the full growth of real estate prices [7,9–12], but it is hard to unravel the mechanism accounting for the failure of effects by the short-term control policy according to current evidence. Filling this knowledge gap is particularly important for decision-making departments to respond to the central government's policy on the real estate market. Studies suggest that current policies may have effects with lags on the relationship between supply and demand [7], which results in short-term failure [9–12]. This paper argues that this failure comes not only from the lag of policy, but is also the reason for short-term policy failure [13].

It was emphasized at the 2016 Chinese Government Economic Work Conference that "houses are for living in, not for speculation" and the government should focus on "promoting the stable and healthy development of the real estate market" [14]. Since 2019, public recognition of real estate prices has been largely disturbed by dual perceptions of the COVID epidemic and Sino-US trade frictions. Entropies started to increase and accumulate in synchronization of disorders of the housing market. Therefore, Chinese local governments released a new round of policies in an attempt to stabilize the market by enforcing short-term regulations. This evoked positive responses but negative associations of large downward pressure on real estate prices. The responsive decline of housing prices up to billions of dollars was described as a "thunderstorm". At the 2021 China Central Economic Work Conference, it was emphasized about the magnitude of "strengthening the guidance of expectations and exploring new development models" [15].

Apparently, decision-makers may intend to explore new dependencies on which the measures can be available to stabilize housing prices. Sentiment is defined as the market's collective beliefs and expectations, and expectation is an important component of sentiment. Hence, the consumer expectation about the housing prices can be reflected by public sentiment. The promulgation of policies, especially those expecting short-term feedback, will cause significant fluctuations in the sentiments of real estate consumers. Therefore, a further question arises about how the specific effects of the implementation of the policy on the sentiments of real estate consumers have been affected. Consumer sentiment may have had a key role in the failure of short-term control policies, but relevant evidence is scarce. The research objective of this study is to establish the role of housing sentiment in policymaking to regulate and stabilize real estate prices.

China's real estate market is dominated by household purchases, which is similar to the stock market with a large number of retail investors. There are spillover effects between cities that received real estate restrictions and that did not. When investments in real estate of local consumers are severely restricted, their demands will be suppressed accordingly and transferred to a neighboring real estate market. Therefore, real estate sentiments in neighboring cities will be stimulated, followed by the increase in local housing prices [16]. Most of the real estate policies are launched firstly on real estate markets that have higher investment values, while the investment values are relatively lower in the neighboring cities where policies have not yet been implemented. It was asserted that urban real estate supply markets have spatial linkages among neighboring cities [17]. When housing trades in neighboring cities are disclosed with relatively low linkage prices, the low investment value will rise rapidly, which will attract local consumers' attention. Local consumers who have expectations for local real estate prices with high investment values will also be buoyed by the impact of the proximity to the market following the introduction of property curbs (Figure 1).

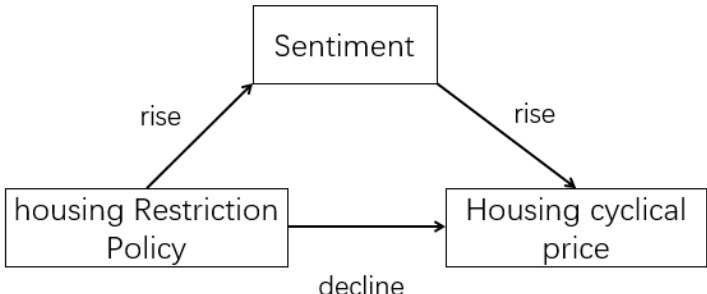

**Figure 1.** Logical Structure of Policy Sentiment Prices.

It is important to figure out a precise approach to evaluate customers' sentiments. The research objective of this study involves optimizing the instrument for assessing housing sentiments. Sentiment is an implicit behavior that is difficult to be measured as an assessment of changes in marketing. The housing sentiment can be estimated through

modeling regression, such as the employment of state-space model. The Markov switching variation (MSVAR) model has been used to analyze the relationships among the effects of short-term regulation policy, housing sentiment feedback, and the cyclical price of real estate. This paper takes 30 large and medium-sized cities in China (This paper selects 30 large and medium-sized cities as samples to construct a short-term regulatory policy effect index. Beijing, Shanghai, Guangzhou, Shenzhen, Tianjin, Hangzhou, Nanjing, Wuhan, Nanchang, Chengdu, Qingdao, Suzhou, Fuzhou, Xiamen, Changsha, Harbin, Changchun, Wuxi, Dongguan, Kunming, Shijiazhuang, Huizhou, Baotou, Yangzhou, Anqing, Yueyang, Shaoguan, Nanning, Lanzhou, Jiangyin) as an example to analyze the impact of the overall short-term market regulation policy on the overall sentiment of real estate. Empirical studies verified that the short-term regulation policy of real estate trades was "tightened", generating stress on cyclical housing prices. Hence, the short-term control policy does contribute to restraining high housing prices, which accounts for the tightening consequence of short-term policy regulation on real estate and an upregulation of positive consumer sentiment. As a consequence, higher positive sentiments among customers would further result in a positive effect on market cyclical prices. In the situation when public sentiments of customers are generally negative, short-term control policies will cause fluctuations in consumer sentiment and reduce the effectiveness of short-term policies by causing partial failure to ensure stable prices.

This article undertakes a comprehensive analysis of short-term control policies, with purchase restrictions serving as the quintessential exemplar. Presently, countries including China, Singapore, Australia, Malaysia, and the United Kingdom have either implemented or previously employed real estate purchase restrictions. Among these nations, China emerges as a paradigmatic case study. Since 2010, the Chinese government has proactively implemented a multifaceted suite of measures to curb excessive property transactions, inevitably accompanied by periodic episodes of tightening and subsequent relaxation. However, while few other countries have embarked upon a comparably protracted voyage characterized by both robust temporal continuity and resolute intensity (as evident in unwavering purchase restrictions), China unmistakably occupies a distinct position in implementing and iteratively adjusting these policies. It is precisely due to these characteristic hallmarks that this article positions China as an exemplary research archetype, aspiring to furnish experiential insights to other nations contemplating the adoption or refinement of their own purchase restriction policies. The paper structure diagram of this article can be found in the Supplementary Materials (Figure S1).

## 2. Study Purposes

Two purposes were set in this study:

(1)　Being taken as a model study that can help optimize the instrument for assessing housing sentiments.

The measurement of emotions poses challenges, yet emotion measurement forms the foundation of behavioral finance analysis. Previous sentiment indices based on principal component analysis (PCA) may suffer from information loss, failing to accurately reflect the true market conditions. Therefore, this study aims to enhance the measurement of emotions by employing a state-space modeling approach.

(2)　To establish the role of housing sentiment in policymaking to regulate and stabilize real estate prices.

Short-term regulatory policies are established to stabilize real estate prices. However, there are situations where policies fail to achieve price stability in the real estate market. This study explores the factors influencing short-term regulatory policies. Consumer sentiment is often regarded as a factor influencing real estate prices. Therefore, investigating the role of housing sentiment in real estate policy formulation not only helps the government to comprehensively evaluate the effectiveness of real estate policies but also enables policy optimization.

## 3. Literature Review

### 3.1. Short-Term Control Policy

China launched a policy of tightening control on real estate in two stages: 2010–2013 and 2016–2021. The expansionary policy adjustments were implemented during 2013–2016. The policy was deployed by the central government of China and formulated in detailed rules according to the national standard, which was documented as a non-mandatory policy [18]. The central government was concerned with the issue of stabilizing housing prices, which was responded to by regional governments alternatively as concerns of growth promotion on housing prices and regional economic development [19]. The policy specifically includes controlling terms, such as restrictions in purchases, sales, down payment proportion, and interest rate for provident fund loans.

Consumers believe that short-term real estate control measures should not be employed following a long-term policy. After the control measures are terminated, real estate prices will rise significantly [18]. A city's administrative level (top-down pressure) is the main driver of real estate restriction policies. Under invisible coercion enforced by higher-level governments, local governments are forced to implement seemingly "non-compulsory" housing policies [18]. Local governments still have independent decision-making and controlling powers over the specific implementation terms of restrictive policies. Once the central government weakens the "implicit" coercion, policies launched by local governments will be canceled or at least weakened by restrictive policies. The Chinese central government has strengthened the implementation of the real estate short-term control policy three times in 2010, 2014, and 2016. The duration of most Chinese city restriction policies was between two and four years. Accordingly, consumers would expect the lifting of restrictions in the future, which are more inclined to house sales during the policy period. At the same time, real estates are also likely to be cashed out when the policy is launched. Therefore, consumers will have higher sentiments after the introduction of the restriction policy.

With the rapid rise of real estate prices, scholars have questioned whether the short-term control policy could validate restriction [1–5], but more voices still affirmed the ability of the short-term control policy to stabilize real estate prices [9–12]. Short-term control policies were released on real estate short-term control policies in the United States and Singapore, both of which succeeded in limiting the excessive rises in prices of real estate sales and rentals [6]. In China, it was also disclosed that short-term control measures had an expected effect on the rate of increasing prices for second-hand house sales and rentals in Beijing [4]. Efficient regulatory policies exert their effectiveness through the employment of joint actions of supply and demand tradeoffs. On the supply side, the post-control short-term policies are represented by restrictions in both purchase and sale. Thereafter, the main impact on the real estate supply and demand in the market has shifted from the demand side to the supply side [8].

At the same time, partial failure and negative effects also emerged following the short-term control policies involving the control of real estate prices [20]. The inconsistency between the goals of the central and local governments has caused a deviation or even a reduction in the employment of controlling effect [21]. This leads to a further issue that policies cannot be conducive to the welfare of people with essential housing needs [22]. As the major operator of housing sale, real estate companies would suffer a decline in sales amount, which makes their commercial performance deeply influenced [23].

At present, the short-term regulatory policymakers mainly focus on the regulation of stability of real estate supply and demand as well as any negative impacts caused by issued regulations. When the housing market is full of customers with high positive sentiments, any type of restrictive policies cannot effectively limit the rise of real estate [13]. However, these restrictive policies may be the driver of rising positive sentiments. It has been confirmed that sentiments can significantly affect urban real estate prices [24], which suggests that it is meaningful to intervene in real estate prices by inducing market sentiments. Short-term control policies can transfer consumers' attention, and meanwhile, real estate



control policies will increase the sentiment of surrounding cities [16]. The emotional shock caused by the perception of short-term regulations will be passed to real estate markets as a driver of lowered prices and unstable housing prices. All abovementioned relationships, however, are still insufficient to predict the direct impact of short-term regulatory policies on consumer sentiment.

### 3.2. Housing Sentiment

In financial markets, sentiment is defined as the collective beliefs and expectations towards the probability of future markets. However, it is a little surprising that this definition has never been fully verified [25]. Sentiment can also be interpreted as the belief itself or the degrees of optimism or pessimism that consumers can perceive about the next-step occurrence in the future [26]. Sentiment is "a belief about future cash flows and investment risk that is not justified by the facts at hand" [27].

Subjective sentiment can enforce an invisible impact on asset prices. For example, investors' emotions can change their decisions regarding buying or selling, which further affects stock prices [25,27,28] and investors' next-step behaviors [29–33]. The real estate market has more individual consumers, who are more likely to generate casual restrictions on transactions and are also more susceptible to immediately perceived emotions compared to collective buyers [32,34–36]. It was suggested that positive sentiments can significantly increase real estate prices. Consumer sentiment occupies a dominant position in the determination of short-term prices, and sentiment has a greater impact in regions with more developed economic levels [37]. When mentioning gains from housing sales and rentals, higher positive sentiment is often accompanied by lower housing gains [13,33,38,39]. Researchers engaging in studies on emotions mainly focus on their relationship with real estate prices. For them, the most interesting topic is how emotions affect real estate prices under the short-term regulation, of real estate is still relatively rare. The government wants to use short-term regulation measures to stabilize market prices, which are sensitive to consumer sentiments.

### 3.3. The Measurement of Housing Sentiment Index

How to measure the sentiment of housing customers remains a challenge. Sentiment represents the difference between asset prices and fundamentals, which are difficult to be assessed through any reliable measurements [40,41]. At present, three major methods are widely considered for quantifying emotional measurement. The first is the proxy index method, which selects market variables as emotional agents, eliminates the influence of fundamentals, and reduces the dimensionality of variables [25,42]. A selection of multiple proxy variables can reflect several market sentiments and eliminate side effects from fundamental factors. A sentiment indicator can be synthesized using principal component analysis as a representative dimensionality reduction tool.

The second is the survey data method, which collects self-reported scores on questionnaires, that are finished by respondents. To assess customer sentiments in a real estate market, it needs a large number of respondents, who can expose their self-perceived sentiments [43]. However, this method is taken to need heavy inputs in expenditures in time and recruitment. Furthermore, the subjective human bias of respondents during the survey cannot be fully eliminated, which affects the matching accuracy of emotional feedback from intended emotions [25,44].

The third is any approaches that employ big data as the source of data for further analysis, such as semantic text analysis. This method heavily depends on online big data crawling and machine learning technologies. Through web crawler technology, the collection of netizens' speeches and facial expressions can be used for measuring emotional responses based on text analysis [45]. However, nearly 80% of Chinese netizens are under the age of 40, and middle-aged people with strong real estate purchasing power account for only 15% [46]. Big data on social networks have accurate feedback on young people's emotions in spite of inevitable deviations.

Therefore, in this paper, the first emotion proxy index method was employed by selecting the proxy indexes of real estate sentiments and uses of the state space model to solve the hidden state variables to analyze the real estate market sentiment.

This paper applies a state-space model to analyze consumer sentiment in the real estate market and study the relationship between real estate short-term regulation policy and Housing Sentiment. It mainly involves two issues: first, whether short-term real estate regulation can affect consumer sentiment; second, whether Housing Sentiment invalidates short-term regulation policy.

## 4. Research Hypotheses

Short-term regulatory policies are inherently temporary and easily subject to changes in the short term. Consumers are aware of this and do not perceive short-term regulation as something that can exist in the long term. Although there may be transaction restrictions in the short term, once regulatory policies are relaxed in the long term, the transaction restrictions on real estate holdings held by consumers will be lifted. Therefore, after a period of tight regulation, consumers tend to have higher expectations of acquiring real estate, while after a period of relaxed regulation, they seek to profit from selling at higher prices. Consequently, the sentiment among real estate consumers significantly improves following short-term regulatory policies. Based on these assumptions, hypotheses were developed:

**Hypothesis 1.** *Short-term real estate regulation policies will improve positive housing sentiment.*

This is disclosed based on previous findings that investors are more susceptible to emotions towards real estate sale market state as well [36], and inducing more consumer sentiments, even under the effect of tightening regulatory policies, will benefit the rise of real estate price and consumers' high expectation [13].

Heightened sentiments contribute to an increase in real estate prices. Real estate consumers experiencing heightened sentiments have higher expectations for real estate prices. These expectations, in turn, drive up the prices in the real estate market. While short-term regulatory policies may affect sentiments and prices, they do not alter the impact of sentiments on prices. Heightened sentiments will continue to be associated with price increases. Based on these assumptions, hypotheses were developed:

**Hypothesis 2.** *Under the short-term real estate regulation policy, real estate prices and housing sentiment will change with the same synchronization.*

Such a potentially unrealistic assumption arises from further speculation on the basis of the first hypothesis.

## 5. Methodology

### 5.1. The Construction of Real Estate Market Sentiment Index

Emotion contains psychological factors during the decision-making process. Since psychological factors are unobservable, various emotional proxy indicators are introduced to assess emotions directly or indirectly [42]. State-space models are widely used to measure unobservable variables. Four emotional index agents will be selected in this paper, and their numeric values will be collected through the self-reported feedback on questionnaires put forth by the People's Bank of China. Consumer sentiments will be analyzed by the state space model regression.

### 5.2. Sentiment Proxy Selection

Four emotional proxy variables were employed in this paper. The first three were selected according to variables used by Das et al. [42], who assessed sentiments through the consumer confidence survey questionnaire that was designed by scholars from the University of Michigan. Proxies quantifying the US housing sentiments were employed as

the percentage of "thinking now is a good time to buy" and the difference in the percent scores between responses to "thinking now is a good time to buy for investment" and "thinking now is a bad time to buy for investment". Similar questions were also employed for respondents on the questionnaires developed by the People's Bank of China. On it, questions included whether "expected an increase in housing expenditure in the next three months" and whether "expected future increase in real estate prices". It was also selected based on the difference in the scores in responses to "expected future increase in real estate prices" and "expected future decline in real estate prices" as the proxy variables.

The real estate entrepreneur confidence index (expectation index) was also chosen as the fourth emotion proxy. The real estate entrepreneur confidence index takes 100 as the critical value, with values changing in a range between 0 and 200. When the confidence index is higher than 100, it indicates that entrepreneurs' development is expected to be bullish in the real estate industry. Conversely, when the confidence index declines to be lower than 100, it was indicated that entrepreneurs' development is evaluated to be bearish in the real estate industry. The sentiments of staff in real estate companies play an important role in real estate sentiment, but the sentiments of builders have a higher impact on market prices than buyers [33]. Entrepreneurs are also important participants in the real estate market. When analyzing real estate sentiment, it is meaningful to include assessments on the sentiments of real estate entrepreneurs in sentiment analysis. The sentiment of real estate entrepreneurs is an important indicator reflecting entrepreneurs' feelings and confidence towards perceptions of the macroeconomic environment. Relevant results can be used to predict the changing trend of economic development.

### 5.3. State-Space Model

This paper will apply a state-space model to measure consumer sentiment in the real estate market. Sentiment proxy indicators all represent a kind of Housing Sentiment, and the implicit unmeasured state variable is sentiment. The measurement equation is, and the state equation is:

$$Y_{it} = Z_i \times (\alpha)_t + e_t \tag{1}$$

$$(\alpha)_t = (\alpha)_{t-1} + \varphi_t \tag{2}$$

where, $Y_{it}$ is the emotion proxy variable; $Z_i$ is the measurement marix; $e_t$ and $\varphi_t$ are white noises of the measurement equation and the state equation, respectively. They contribute to the requirement of normal distribution of data with zero means and constant variance; $(\alpha)_t$ indicates the hidden unobservable variable Housing Sentiment at time $t$.

### 5.4. Real Estate Sentiment Index

Sentiment does not include the influence of fundamental factors [27]. This article refers to Das et al. [42] macroeconomic choice and returns a group of related macroeconomic fundamental indicators, excluding economic fundamental factors. Apply the following variables to represent fundamentals: nominal gross domestic product (GDP), nominal per capita income (incpc), unemployment rate (ur), nominal mortgage rate (mr), inflation (CPI), nominal industrial production (ip), and nominal money supply Quantity (M1)

$$PROXY_{it} = \alpha_i + \sum_{i=1}^{n} \beta_i FUND_{it} + \varepsilon_{it} \tag{3}$$

where $PROXY_{it}$ represents the $i$-th original emotional variable at time $t$, which captures rational expectations and emotions at the same time as a constant in the model; $FUND_{it}$ represents a collection of macro-economic fundamentals; $\beta_i$ indicates that the dummy variable controls the seasonality factor; residuals $\varepsilon$ represent irrational, purely emotional proxies. The four pure emotional proxy variables were input to Equations (1) and (2) with the Kalman filter together, resulting in the filter of the state variables. Dynamic changes in the housing sentiment index are shown in Figure 2.

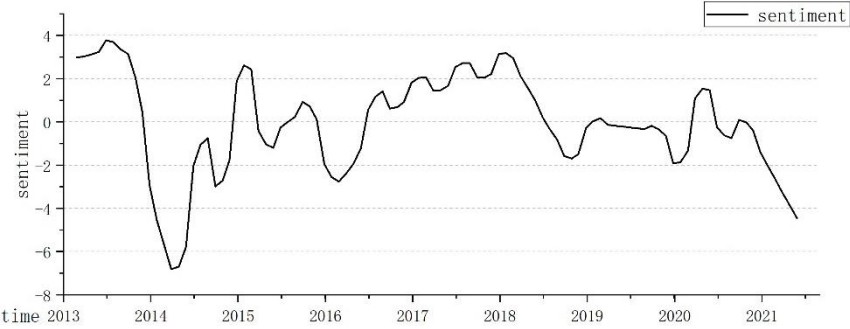

**Figure 2.** Yearly dynamic changes in housing Sentiment index from 2013 to 2021.

*5.5. Comparison of Sentiment Indexes*

In order to verify the rationality of the sentiment indexes, they were generated by the state-space model with dependent input of values from principal component analysis. The VAR model was used to verify the relationship between sentiment and price, according to Hui et al. [47]. Both indices were found to be useful for predicting housing prices. The comparison results of impulse responses are shown in Figure 3. The coefficient Senti.price is the impulse response of the state-space model sentiment index to price. Both coefficients of senti1.price and senti2.price present the promotion of positive sentiment on the increase in the first period and subsequent decrease, respectively. This is consistent with the findings of Hui et al. [47]. The AIC (Akaike information criterion) and BIC (Bayesian Information Criterion) of the state space model sentiment index are −1435.213 and −1326.218, respectively. The eigenvalues of AIC and BIC in the principal component analysis are −1434.235 and −1325.24, respectively. Values of AIC and BIC for sentiment indices of the state-space model are significantly lower than those indicated by principal component analysis. This indicates that the sentiment index generated by the state-space model has a higher accuracy than the sentiment index generated by the principal component analysis for predicting prices.

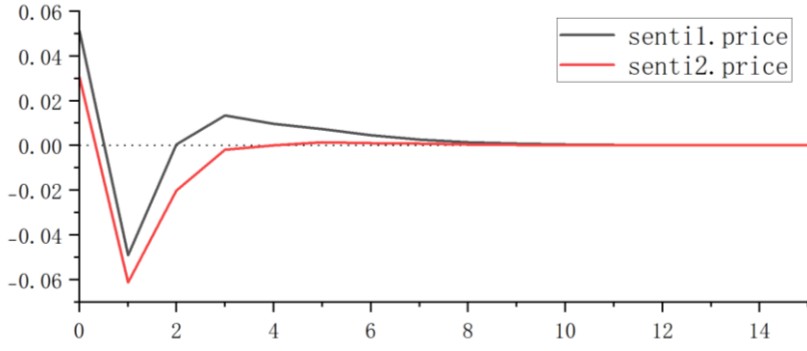

**Figure 3.** Comparison results of impulse response function.

## 6. Data

The empirical research investigation focuses on exploring the intricate relationship between short-term regulatory policies, real estate sentiment, and cyclical price patterns. Specifically, it centers its attention upon a meticulously selected sample encompassing thirty major and mid-sized cities across China. The short-term regulatory policies are operationalized through the acquisition of policy collection indices pertinent to these thirty cities. Concurrently, real estate sentiment is probed via surveys conducted by the central bank, serving as a suitable proxy to gauge investor sentiment within the Chinese real estate market. The paramount objective of this study lies in deciphering the transient fluctuations exhibited in real estate prices. Thus, to achieve this aim effectively, the Hamil-ton-HP filter methodology is adroitly employed to discernibly extract and eliminate any long-term

trends, thus enabling an intensified examination of the underlying short-term fluctuations characterizing real estate prices.

The fundamental data consist of average prices of new residential properties in 30 large and medium-sized cities (excluding second-hand housing transaction prices) sourced from the Wind database. The Wind database is a comprehensive financial database developed and maintained by Wind Information Company Limited, Shanghai, China, serving as a platform for investors and financial institutions.

In this paper, the sample time span is established by selecting monthly data from June 2013 to September 2021. Specific variables include cyclical real estate prices, real estate sentiment, real estate transaction volume, policy index, and loan interest rate and money supply (M2). Consumer sentiment is evaluated by Formula (2).

### 6.1. Periodic Real Estate Price

In this paper, the average prices of new residential properties are selected as a variable, which is obtained by extracting and processing the average price of real estate in 30 large and medium-sized cities. China has been in a state of rapid development for decades, during which real estate prices showed a long-term upward trend. When measuring the impact of short-term regulatory policies on prices, full consideration must be given to eliminating the impact of long-term trends. The Hamilton-HP filter was used to remove the influence of long-term trends and obtain periodic housing prices. Hamilton [48] modified the HP filter method and ensured that the identified residual components were stationary. It provided consistent estimates for various unknown and possibly non-stationary processes. As suggested by Hamilton [48], two years are used as the standard base period. Dynamic changes in real estate cyclical prices are shown in Figure 4.

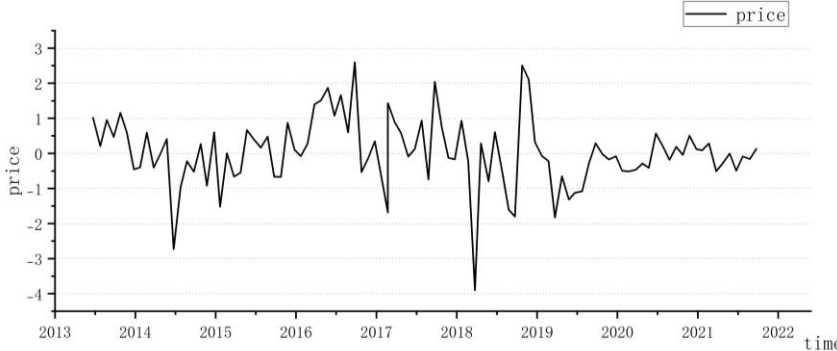

**Figure 4.** Dynamic changes in real estate cyclical prices.

### 6.2. Short-Term Control Policy Index

The policy effect index reflects the magnitude of short-term control policies. The policy effect thermometer method was used to measure the degree of regulatory policies. This method can artificially assign weights to distinguish the strength of different policies. First of all, the strength of the short-term regulatory policies issued by each city in the current month is scored using a criterion in Table 1. The policy data in this article were derived from the China Real Estate Information Policy Database and collected manually by the authors.

**Table 1.** Scoring criterion of policy effect.

| Direction of Purchase Restrictions | National Policy | Down Payment Ratio | Household Registration Restrictions | Different Regions | Tax Conditions | Others | Sum |
|---|---|---|---|---|---|---|---|
| restriction | 2 | 2 | 1 | 1 | 1 | 1 | |
| remove restrictions | −2 | −2 | −1 | −1 | −1 | −1 | |
| unpracticed | 0 | 0 | 0 | 0 | 0 | 0 | |

According to the scoring standards shown in Table 1, the policy effects of all cities in the current month are scored separately to obtain the regulatory policy effects in each

city. The policy index is the average value of the policy effects of all investigated cities, denoted by n, which is the surveyed number of samples. This paper selects 30 large and medium-sized cities as samples to construct a short-term regulatory policy effect index: Beijing, Shanghai, Guangzhou, Shenzhen, Tianjin, Hangzhou, Nanjing, Wuhan, Nanchang, Chengdu, Qingdao, Suzhou, Fuzhou, Xiamen, Changsha, Harbin, Changchun, Wuxi, Dongguan, Kunming, Shijiazhuang, Huizhou, Baotou, Yangzhou, Anqing, Yueyang, Shaoguan, Nanning, Lanzhou, Jiangyin. Add up with the previous month's index according to an equation:

$$PE'_{t+1} = \frac{1}{n} \sum_{i=1}^{30} \alpha_i + PE'_t \tag{4}$$

The examples of Real Estate Policies are presented in Table 2.

**Table 2.** Examples of Real Estate Policies.

| City | Date | File Name | Content | Rating |
|------|------|-----------|---------|--------|
| Shanghai | 6 December 2013 | Notice of Shanghai Municipality on Relevant Issues Concerning the Strict Implementation of Housing Purchase Restriction Measures | Down payment ratio + household registration | +3 |
| Beijing | 14 August 2015 | Notice of the Beijing Municipal Commission of Housing and Urban-Rural Development and the People's Government of Tongzhou District on Strengthening the Management of Commodity Housing Sales in Tongzhou District | Household registration | +2 |
| Tianjin | 30 September 2016 | Implementation Opinions of the General Office of Tianjin Municipal People's Government on Further Promoting the Steady and Healthy Development of the Real Estate Market in Our City | Household registration + down payment | +4 |
| Xiamen | 25 August 2014 | Implementation Opinions of Xiamen Land Resources and Real Estate Administration and Other Departments on Promoting the Steady and Healthy Development of the Real Estate Market | Household registration | +2 |
| Shijiazhuang | 25 September 2014 | Notice of Shijiazhuang City Housing Security and Real Estate Administration Bureau on Canceling the City's Housing Purchase Restriction Policy | Cancel household registration | −2 |

### 6.3. Additional Variables

The transaction quantities were employed as variables. The transaction volume was quantified as the monthly data on the number of commercial housing transactions in 30 large and medium-sized cities. In this paper, the monthly average exchange rate of RMB is selected as a variable, and the monthly average of M2 is used as the money supply. (The above data come from the Wind database. The policy data come from the policy database of Guoxin Real Estate Information Network, and this paper organizes it manually. The Guosen Real Estate Information Network database is sponsored by the China Information Center, which is China's national information and data center).

The Markov switch VAR requires that the time series should be stationary, considering that variables cannot be regressed in different dimensions and in the economic sense. Therefore, the first-order logarithmic difference processing is performed on the non-stationary variables. The descriptive statistics of specific variables can be seen in Table 3.

**Table 3.** Descriptive statistics of variables.

|  | N | Average Number | Maximum | Minimum | Standard Deviation |
|---|---|---|---|---|---|
| Sentiment | 100 | −0.05941 | 3.78835 | −0.06549 | 2.31665 |
| Ln Pri | 100 | 0.0242 | 2.60183 | −0.02191 | 0.98786 |
| ΔlnAcr | 100 | $7.02 \times 10^{-4}$ | 0.0834 | 0.0011 | 0.02245 |
| lnNum | 100 | 11.93477 | 12.5198 | 11.96305 | 0.32219 |
| ΔlnRate | 100 | $4.13 \times 10^{-4}$ | 0.0377 | $-3 \times 10^{-4}$ | 0.00989 |
| ΔLnM2 | 100 | 0.0081 | 0.0252 | 0.00765 | 0.00798 |
| ΔLngover | 100 | 0.02162 | 0.79048 | 0 | 0.09937 |

Within the realm of data descriptive statistics, this article incorporates elements of model assumptions. The foundational postulates of the Msvar model entail variables that conform to a transition probability matrix and shared structural constraints. At its core, the model presupposes a robust gradient of transformation between these variables. Simultaneously, the key variables adhere to autoregressive model assumptions, featuring inherent time dynamics. Stationarity tests serve as a prerequisite for the variables to meet the model's requirements. Considering the policy index data, it is evident that a conspicuous gradient exists in response to recurrent instances of policy tightening and loosening, with associated policy effects displaying temporal delays that align with the model's fundamental assumptions. Furthermore, real estate sentiment exhibits two distinct gradients–exuberance and despondency–along with pronounced temporal patterns. Consequently, the MSVAR model assumptions are aptly fulfilled.

## 7. Empirical Research Model

### 7.1. Markov Switching VAR Model

Krolzig [49] combined the VAR model with the Markov Switching model to obtain the MSVAR model, which can assume that the parameters change with the transformation of the economic system. Different from the traditional VAR model, this model can involve the nonlinear characteristics of macroeconomic variables. The short-term control policy of China's real estate is switched between the tightening and loosening of the two policy states. The MSVAR model can be used to regress this policy transition. The number of states in the MS method has been set prior to the model equation, which can identify the time and probability of occurrence of different economic states.

Assuming that there are M regimes and T periods in the MSVAR model, for the p-order autoregressive of the K-dimensional time series vector, the expression of the intercept term is:

$$y_t = \mu(s_t) + A(s_t)(y_{t-1}) + \cdots + A_p(s_t)(y_{t-p}) + \varepsilon_t$$
$$\varepsilon_t \sim NID[0, \textstyle\sum(s_t)] \tag{5}$$

The expression for the change in mean:

$$y_t - \mu(s_t) = A_1(s_t)(y_{t-1} - \mu(S_{t-1})) + \cdots + A_p(s_t)(y_{t-p} - \mu(s_{t-p})) + C(s_t)Z_t + \varepsilon_t \tag{6}$$

where $\mu(s_t)$, $\mu(s_{t-1})$, ..., $\mu(s_{t-p})$ are the average parameters associated with the state variables. State system transition probability can be calculated as:

$$P_{ij} = Pr(s_{t+1} = j, s_t = i), \sum P_{ij} = 1, \forall i, j \epsilon(1, \cdots, M) \tag{7}$$

Subsequently, the probability matrix can be described as:

$$P = \begin{pmatrix} P_{11} & \cdots & P_{1M} \\ \vdots & \ddots & \vdots \\ P_{M1} & \cdots & P_{MM} \end{pmatrix} \tag{8}$$

where the total sum of $\forall i \epsilon (1, \cdots, M), P_{i1} + P_{i2} + \cdots + P_{iM}$ is 1.0.

In the MSVAR model, it is not assumed that all parameters are related to the state variables in practical applications. For simple processing, specific parameter settings are usually associated with the state variables. Variables in mean, intercept, coefficient, and variance vary with the state variables, thereafter, different MSVAR models can be established. The classification of MSVAR models can be seen in Table 4. Specific models are established according to AIC, SC, HQ criteria.

**Table 4.** Classification of MSVAR models.

| Variables | | MSM | | MSI | |
|---|---|---|---|---|---|
| | | μvarying | μinvariant | μvarying | μinvariant |
| $A_j$ varying | $\sum$ varying | MSMAH-VAR | MSAH-VAR | MSIAH-VAR | MSAH-VAR |
| | $\sum$ invariant | MSMA-VAR | MSA-VAR | MSA-VAR | MSA-VAR |
| $A_j$ invariant | $\sum$ varying | MSMH-VAR | MSH-VAR | MSIH-VAR | MSH-VAR |
| | $\sum$ invariant | MSM-VAR | Linear VAR | MSI-VAR | Linear VAR |

### 7.2. Lag Order, Regime, and Model Determination

Before establishing the MSVAR model, it is necessary to determine the lag order and state numbers. Model form is selected according to the criteria of AIC, HQ, and SC. Since the lag order is longer than expected, the higher degrees of freedom are lost. Variables of AIC, BIC, SC, and FPE were used for testing, among which, the first-order lag is $-0.0479$ (AIC), $-0.0186$ (BIC), $0.0245$ (SC), and $0.9532$ (FPE). The four testing results all show that the first-order lag is the best. Therefore, the first-order lag was chosen in this study. The choice of the number of district systems is mainly determined by the actual situation of the research question. We divide states of the real estate market into two types: policy tightening and policy relaxation. Therefore, this paper sets the number of district systems to 2. The classification of the MSVAR model is divided into types of mean and intercept. The numerical results of AIC, SC, and HQ are shown in Table 5.

**Table 5.** Model Selection and log-likelihood results for AIC, HQ, and SC.

| | | Log-likelihood | AIC | HQ | SC |
|---|---|---|---|---|---|
| MSM | MSM(2)-VAR(1) | 612.6387 | $-10.9422$ | $-10.1892$ | $-9.081$ |
| | MSMA(2)-VAR(1) | 430.099 | $-6.5273$ | $-5.3924$ | $-3.7224$ |
| | MSMH(2)-VAR(1) | 681.898 | $-11.9171$ | $-10.9414$ | $-9.5055$ |
| | MSMAH(2)-VAR(1) | 430.099 | $-6.103$ | $-4.7454$ | $-2.7477$ |
| MSI | MSI(2)-VAR(1) | 595.5026 | $-10.596$ | $-9.843$ | $-8.7349$ |
| | MSIA(2)-VAR(1) | 702.9927 | $-12.0403$ | $-10.9054$ | $-9.2354$ |
| | MSIAH(2)-VAR(1) | 695.0783 | $-11.4561$ | $-10.0986$ | $-8.1008$ |
| | MSIH(2)-VAR(1) | 713.235 | $-12.5502$ | $-11.5745$ | $-10.1386$ |

From the criteria of AIC, HQ, and SC, it can be seen that MSIH (2)-VAR (1), the model in the mean model is better than other models in log-likelihood, AIC, HQ, and SC. Hence, the equation of MSIH (2)-VAR (1) was used. The estimated results of the model are the basis for the analysis.

## 8. Results

### 8.1. Model Estimation Parameter Results

The results of parameter estimation are shown in Table 6. Therein, Const represents the size of variables under different regional systems. SE represents the volatility of variables

under different regional systems. There are significant differences between district regimes-1 and -2. In real estate cyclical price (pri), price of district Regime 1 (−9.8027) is lower than that of district Regime I (−9.4069), but the change of district Regime 1 (0.9171) is higher than that of district Regime II (0.8749). The emotion of district Regime 1 (−1.001) was lower than that of district Regime 2 (−0.6789). The change of district Regime 1 (0.3363) was significantly smaller than that of district Regime 2 (0.4835). The emotion of district Regime 2 was higher and the change was greater. In terms of policy regression, the policy of district Regime I (−0.0075) is smaller than that of district Regime II (0.0582), and the policy of district Regime I is more relaxed. The characteristics of Regime1 (low price, large price fluctuation, low sentiment, low sentiment fluctuation, loose policy), and the characteristics of Regime 2 (high price, small price fluctuation, high sentiment, high sentiment fluctuation, policy tightening).

**Table 6.** MSIH (2)-VAR (1) model coefficients.

|  | **senti** | **pri** | **num** | **rate** | **M2** | **gover** |
|---|---|---|---|---|---|---|
| Const (Reg. 1) | −1.001 | −9.8027 | 7.4691 | 0.0332 | 0.0517 | −0.0075 |
|  | (−0.7264) | (−2.6658) | (9.1429) | (1.2492) | (1.5547) | (−0.1115) |
| Const (Reg. 2) | −0.6789 | −9.4069 | 7.5705 | 0.0415 | 0.0487 | 0.0582 |
|  | (−0.4876) | (0.8749) | (9.1958) | (1.5269) | (1.4509) | (0.7739) |
| SE(Reg. 1) | 0.3363 | 0.9171 | 0.3093 | 0.0061 | 0.0077 | 0.0156 |
| SE(Reg. 2) | 0.4835 | 0.8749 | 0.1070 | 0.0116 | 0.0066 | 0.1686 |

The state probabilities of the MSIH(2)-VAR(1) model are shown in Figure 5. The filtered, predicted, and smoothed probabilities are evaluated for the two regime states. Smoothed probabilities were used to distinguish mechanism states. The judgment method is: If the smoothed probability of a specific mechanism state in a specific month is greater than 0.5, then it is judged to be in the mechanism state. The monthly data selected for this article follow the same rules.

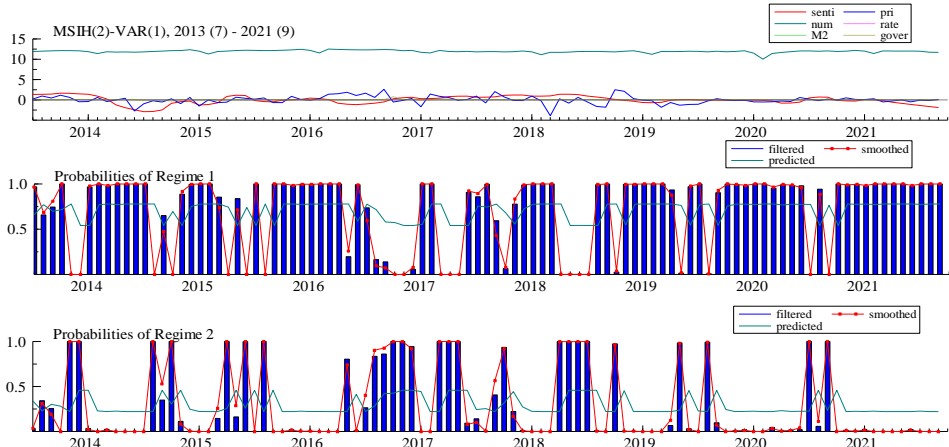

**Figure 5.** MSIH (2)-VAR(1) regional distribution.

Through the above analysis, we can judge that Regime 1 represents a period of stable policies, sentiment, and prices. Regime 2 represents a period of tightening policies, high sentiment, and rising prices. In order to confirm these policy changes, it sorts out the implementation of China's real estate market regulation policies during the sample period. Combined with the policy index calculated in this paper, it is found that district Regime 2 has well identified the policy changes from June 2016 to September 2017 and June 2020

to December 2020. It is worth noting that the Chinese government launched a series of restrictive measures in October 2016. In Figure 5, we find that the second district system has well identified adjustments of this policy. In district Regime 2, the policy index has risen apparently. Regime 2 is the period for the launch of a tightening short-term control policy. Regime 1 is a period of short-term regulation and control policy stabilization or relaxation. In order to confirm this inference, this article sorts out several changes in short-term control policies. Looking at the district Regime distribution map and the real estate market, China's 2014 policy index decreased, and some cities gradually relaxed their control policies. From then until 2016, the city not only canceled regulatory policies but also introduced new restrictive policies. All these were accounted for by the second district Regime during this period. At the end of 2016 and the beginning of 2017, two relatively large restrictions were introduced that covered 20 cities. This period belonged to the second Regime. At the end of 2018, real estate control policies were strengthened.

The temporal scope of this research extends until September 2021. Within the study sample, a distinct period emerged during the COVID-19 crisis, as depicted in Figure 5. This period is characterized by long-term low prices, significant price fluctuations, subdued market sentiment, limited fluctuations in sentiment, and a lenient regulatory environment. This pattern can be attributed to the initial stages of the COVID-19 outbreak, wherein the Chinese government, in response to the unforeseen circumstances, refrained from implementing stringent short-term regulatory measures. Instead, their aim was to mitigate the disruptive impact of the virus on the real estate market. Approximately one year after the onset of the pandemic, the Chinese government's efforts to implement regulatory controls marked a phase of tightened regulations in early 2021 (regime 1). However, as subsequent waves of COVID-19 ensued, the authorities opted not to intensify further regulations in the real estate sector. The model employed in this study effectively captures the process of real estate regulation undertaken by the Chinese government during the initial outbreak of the COVID-19 pandemic. It is noteworthy that the relationship between policies and sentiments during the pandemic aligns with the pre-pandemic era.

Table 7 shows the transition probabilities between the two regimes. According to the model results, a total of 70.1 samples are in district Regime 1, and other 28.9 samples are in district Regime 5. The occurrence probability of district Regime 1 is 0.7099; the average duration is 4.52; the probability of district Regime 2 is 0.2901; the average duration is 1.85. This feature is consistent with the short-term control policy on real estate. During the sample period, the real estate market was implemented to come across short-term regulations and strengthened restrictive measures. The market was transitioned to the second period of district Regime. The influence on emotions gradually weakened over time, the market returned to the period of the first district Regime. Therefore, the real estate market in the first period of district Regime appears longer. The continuous probability of district Regime 1 is 0.7788, and the probability of transition from district Regime 1 to district Regime 2 is 0.2212. The probability of converting district Regime 2 to district Regime 1 is 0.5413, and the probability of district Regime 2 maintaining is 0.4587. This suggests that the market is more likely to be in a state of one.

**Table 7.** MSIH (2)–VAR (1) regional Regime conversion probability.

|  | **Regime 1** | **Regime 2** | **nObs** | **Prob.** | **Duration** |
|---|---|---|---|---|---|
| Regime 1 | 0.7788 | 0.2212 | 70.1 | 0.7099 | 4.52 |
| Regime 2 | 0.5413 | 0.4587 | 28.9 | 0.2901 | 1.85 |

*8.2. Impact of Short-Term Control Policies on Housing Sentiment*

The impulse response was used to analyze the impact of regulatory policies on real estate sentiment in the model. The impulse response can be used to analyze the short-term impact on the relationship between variables in the economic Regime. A standard

deviation impact is imposed on the variables to obtain the policy impulse response under different Regimes.

Figure 6 shows the impact of policy shocks on housing sentiment under the two regimes. In both regimes, sentiment improved significantly following policy shocks. In district Regime 1, in response to the policy shock, the emotional score increased significantly up to a level where the response exceeded 0.01 and reached a plateau in the period of 60. In district Regime 2, following the policy shock, the emotional response increased significantly until the response exceeded 0.1, and the 60 period reached a plateau as well. Regardless of whether it is in district Regime 1 or district Regime 2, in the second period, the policy had a positive impact on sentiment which will continue. The impact of such policy shocks eventually leveled off. Policy tightening has significantly improved Housing Sentiment, verifying Hypothesis 1.

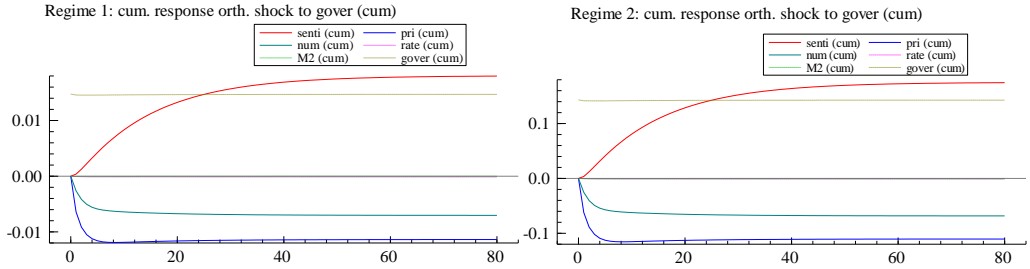

**Figure 6.** Policy Impulse Response (Cumulative Response).

### 8.3. Impact of Short-Term Control Policies on Real Estate Cycle Prices and Real Estate Transaction Volume

Figure 6 shows the impact of policy shocks on transaction volume under the two regimes. After the policy was lifted, the transaction volume decreased significantly. After a one-standard-deviation shock to the policy, in Regime 1, the transaction volume response decreased by an extent of over −0.01 and reached the lowest and stabilized in the 40th period; in Regime 2, the transaction volume response decreased significantly by more than −0.1. In Period 40, it reached a minimum level and kept being stabilized until the end. Afterwards, whether it was the first district Regime or the second district Regime, the response to the transaction volume after the policy shock of the regulatory year dropped significantly, indicating that the tightening of the short-term regulatory policy has significantly reduced the transaction volume.

### 8.4. Influence of Sentiment on Real Estate Cycle Price

Figure 7 shows the impact of sentiment shocks on cyclical prices. In Regime 1, after the sentiment was impacted, the cyclical price rose to reach the highest level in period 34 and subsequently remained stable. In Regime 2, sentiment had a negative impact on price in the first period but immediately changed in the second period, and maintained a positive impact until the impact disappeared. It peaked at 44 and remained stable. After the sentiment shock, real estate cyclical prices rose significantly. Higher sentiment boosted real estate cycle prices significantly, no matter whether during policy was tightened or eased. All these results contribute to the proof of Hypothesis 2.

To sum up, this paper finds that the tightening of short-term control policies has improved the sentiment of real estate consumers. Regardless of whether the short-term control policies are tightened or relaxed, the high sentiment will significantly increase the real estate cycle price. The tightening of short-term control policies has significantly reduced the real estate market price.

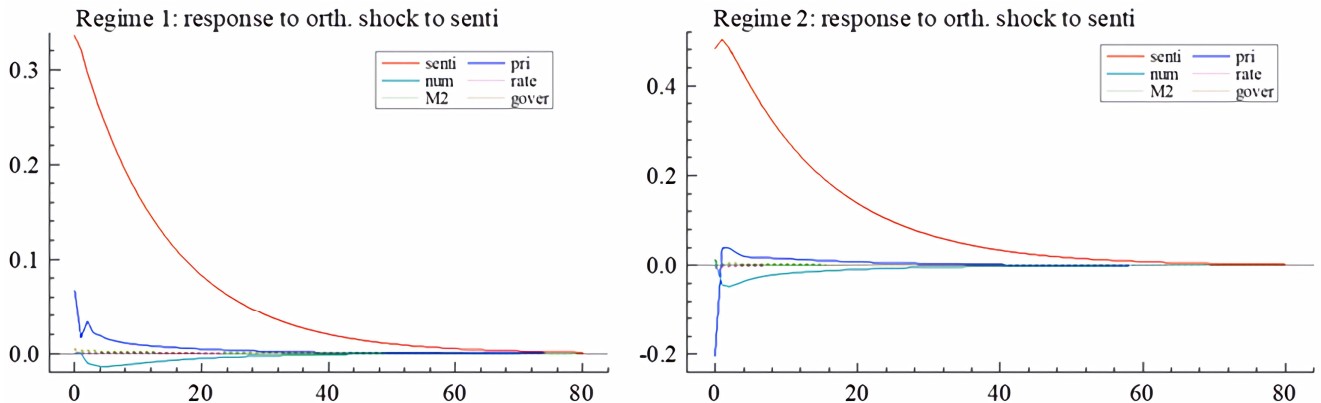

**Figure 7.** Emotional Impulse Response. Abbreviations: senti, Housing Sentiment index; pri, Periodic real estate prices; num, transaction quantity number; rate, rate of RMB; M2, Money Supply M2; gover, Short-term control policy index.

## 9. Discussion

### 9.1. Discussion of Findings of This Study

We found that principal component analysis resulted in a low recovery of information in raw data when being used as an available instrument for assessing sentiment indexes. To cope with this drawback, we employed a new modeling method, namely the state-space model regression, for optimization with higher recovery of raw data information. According to the comparison of these two methods, we also found that the state-space model had a higher prediction precision compared to principal component analysis, which also reflected consumers' expectations. Therefore, our employment of the state-space model can be taken as a recommended instrument for assessing housing sentiments.

Based on results obtained out of state-space modeling, we found that the implementation of short-term regulation policy can be available to prohibit swift increases in real estate prices. As an intervening variable, low positive sentiment may reduce the effectiveness of policy regulation. Our findings generally demonstrated that launching a short-term regulation policy can benefit declines in housing prices, which concur with current results in other studies [1–5]. Our study further extended the exploration of the mechanism and found that, in periods post-launch of short-term regulation policy, positive sentiments of housing consumers were significantly promoted. This verifies and confirms our first hypothesis. Thus, current studies have a rare contribution to the understanding of regulation policy on sentiments [36,37]. In contrast, our study demonstrated that it was the rise in positive sentiments that affected periodic prices of real estate, which also concurs with the confirmation of our second hypothesis. Therefore, the findings of our study suggest that customers' negative sentiments intervened in the short-term effectiveness of regulation policy in the process of price stabilization.

### 9.2. Innovative Novelty Highlight

In this study, we assessed housing sentiment by a novel methodology through modeling. Both economic variables and self-reported scores collected from questionnaires were taken as emotional proxy variables, which were analyzed through principal component analysis to synthesize sentiment indices [27,39,42]. However, the principal component dimensionality reduction method would also cause the loss of original information in emotional agents, which failed to reflect the varied effects of different emotional agents on sentiments. A state-space model and a Kalman filter analysis were used to generate a sentiment index. The state-space model is an important tool for exploring unmeasured variables, but market sentiment is a hidden state variable that is difficult to be observed. This can preserve the original information of proxy variables to the greatest extent of expectation. Our methodology can also improve the accuracy of sentiment measurement.

Former studies mostly attributed the partial failure of short-term regulation on real estate prices to a lagging effect of post-implementation of short-term policies [7]. Our results verified that the partial failure of implementation of short-term control policies was attributed to housing sentiment. Short-term control policies used to be analyzed for the impact on the real estate market from the perspective of elasticities of supply and demand [6–8]. Our study, however, expanded the objective of short-term regulation policy from the perspective of emotion to periodic changes in sentiments on real estate before and after the short-term regulation policy. This was responsible for the novelty of our findings and the reasons for the failure of short-term control policies.

*9.3. Limits of This Study*

Our study has limits. The present research endeavors to scrutinize the real estate market in 30 large and medium-sized cities across China, regrettably lacking a discerning differentiation among these cities. Accordingly, this study falls short in its endeavor to undertake distinct analysis of the prevailing sentiments in each city and explore the pathways and modalities by which diverse municipal policies influence market sentiment. Hence, a further investigation is warranted to probe the potential ramifications of short-term regulatory measures on the real estate sentiments of neighboring vicinities vis-à-vis a singular city's market. Concurrently, while this investigation primarily caters to sizable urban agglomerations, it behooves us to venture into the exploration of whether analogous characteristics pervade the purview of smaller municipalities.

## 10. Conclusions and Suggestions

The short-term control policy can not only reduce housing prices but also increase the sentiment of real estate consumers. The high positive consumer sentiment will increase housing prices, which means that housing sentiment has caused the failure of short-term control policies on price depression. Based on this conclusion, this paper proposes the following policy implications:

It is recommended that governmental authorities curtail the frequency of employing short-term regulatory interventions. The transient nature of such measures tends to attenuate their impact on the sentiment of real estate investors. Excessive employment of such regulatory tactics not only exacerbates investor sentiment but also diminishes the potency of such policies. Consequently, it is advised that the government exercises restraint in relying on short-term regulatory measures and instead prioritizes the establishment of durable mechanisms for real estate control.

Given the transitory nature of investor sentiment, which wanes in influence over time, it is imperative for governments to enact comprehensive policy adjustments promptly when deploying short-term regulatory measures. Acknowledging the delayed onset of policy effectiveness due to the influence of sentiment, it is crucial for governments to refrain from hastily introducing new regulatory policies in response to the initial lackluster outcomes. Engaging in such a course of action would only result in excessive short-term regulation, undermining the desired outcomes.

**Supplementary Materials:** The following supporting information can be downloaded at: https://www.mdpi.com/article/10.3390/su151612660/s1.

**Author Contributions:** Conceptualization, T.Z.; methodology, T.Z.; software, T.Z.; validation, T.Z. and H.G.; formal analysis, T.Z.; investigation, T.Z.; resources, H.G.; data curation, T.Z.; writing—original draft preparation, T.Z.; writing—review and editing, T.Z.; visualization, T.Z.; supervision, H.G.; project administration, H.G.; funding acquisition, H.G. All authors have read and agreed to the published version of the manuscript.

**Funding:** This research received no external funding.

**Institutional Review Board Statement:** Not applicable.

**Informed Consent Statement:** Informed consent was obtained from all subjects involved in the study.

**Data Availability Statement:** Data sharing is not applicable to this article.

**Conflicts of Interest:** The authors declare no conflict of interest.

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
