# Peer review of "State-Space Modeling of Housing Sentiment for Regressing Changes of Real Estate Prices Following Short-Term Control Policy in China"

_sustainability, doi:10.3390/su151612660_

Round 1
Reviewer 1 Report
The topic of this study has strong practical significance, which is helpful to guide the healthy and balanced development of China's real estate. The method of state model used in this paper is appropriate and the correct research conclusion is drawn, which has constructive significance for China's real estate regulation.
This paper discusses the influence mechanism of short-term regulation policy on housing sentiment, which is innovative. Of course, does this study have some limitations and how can it be improved in the future? I hope it will be discussed in the paper.
Finally, the policy recommendations of this paper are not very targeted and need to be further improved.
Reviewer 2 Report
Dear Authors,
Thank you for inviting me to review the paper titled "Short-term Control Policy and Housing Sentiment." While the paper addresses a relevant topic, the abstract and introduction require restructuring for clarity. The researchers needed to clearly state the research problem or aim or explain the study’s rationale and potential applications. It would be helpful to emphasize how the research findings can be utilized. There are several reasons why this research is being rejected in its current form:
First, the title should be more descriptive and provide information on the problem, research goal, field of study, and research methods used. Currently, it is difficult to identify these critical details.
Second, based on my review of the abstract and introduction, I must reject this study due to methodological issues. It does not adhere to the standards of experimental studies and would benefit from a more thorough and scientifically sound approach. The study would need to be rewritten and improved before it can be considered for acceptance.
Third, the researchers needed to clearly state their study's purpose, making it difficult to determine its significance at the international or local scientific research or practice level. Understanding the benefits and potential future studies that can be built upon this research is essential.
Fourth, the research lacks clarity on its intended field. It is unclear if it focuses on housing issues related to laws and policies or on the impact of real estate prices on consumers. It would be beneficial for the researchers to specify the subject and scope of the study within the context of international studies on urban sustainability.
Fifth, to ensure the study's justification, aim, and contribution are clear, it's important to accurately explain the paragraphs that relate to these aspects. Some examples include:
- "However, the existing research fails to tell us why the short-term control policy partially fails, which is particularly important for decision-making departments to respond to the central government's policy on the real estate market."
- Sentiment will influence users' behaviour, although it is difficult to measure.
- The issuance of policies greatly influences consumer sentiment.
Sixth, this research contains several paragraphs that use terms that are not clearly defined, making it difficult to understand their relevance, such as:
- This paper attempts to provide "empirical answers to the above questions".
- "The MSVAR model" is used to analyze the relationship between the short-term regulation policy, Housing Sentiment and the cyclical price of the real estate market.
- This paper takes "large and medium-sized cities in China" as an example to analyze the impact of the overall short-term market regulation policy on the overall sentiment of real estate.
- "Empirical studies" have found that after the short-term regulation of real estate is tightened, it has a significant negative impact on cyclical housing prices.
Seventh, the research would benefit from a more transparent methodology that clearly outlines the research design and methods used. Additionally, a discussion of the findings would be helpful.
Reviewer 3 Report
This paper used the state-space model to analyze Housing Sentiment, and explores the impact mechanism of short-term regulatory policies on Housing Sentiment. However I missed the broader approach of analysed problem. Authors manly focused on Chine case, however what about other countries, because Chine is very specific country and responses can be different. The descriptive data analysis also should be provided the assumsions of the used models. How authors can guarantee the validation of the data? What the impact of Covid-19 pandemic? Furthermore, when we speak about price, about what category of products you are speaking, because in general the results also can be disturbed.
Reviewer 4 Report
1. The paper explores the impact mechanism of short-term regulatory policies in China on housing sentiment. As an example, the Authors look at large and medium-sized cities in China to analyze the impact of the overall short-term market regulation policy on the overall real estate sentiment.
2. The research presented for review is relevant because it helps to understand the impact of government interventions on the housing market and informs policymakers about the effectiveness of these measures in curbing house price growth and maintaining economic stability. It also has broader implications for social impact, investor sentiment, and long-term policy planning within China and in the global context.
3. The Authors were able to prove that short-term control policy will also increase the sentiment of real estate consumers, and the high consumer sentiment will significantly increase the price. The Authors argue that consumer housing sentiment led to the failure of short-term control policies. Such a conclusion suggests that the findings answer the main question posed in the paper.
4. The abstract is structured in accordance with the requirements of the Journal. The hypothesis of the study is given and presented in the paper, which additionally emphasises the scientific significance of the paper. The literature corresponds to the topic of the study. However, there are several weaknesses, namely:
1) Although the text of the paper indicates the methods used in the empirical part of the paper, however, the paper would have benefited if the research methods section had been given separately.
2) It is recommended to expand the keywords to 5-6 keywords.
3) The literature and references to sources are not appropriately formatted.
4) The Authors did not clearly state the limitations of their research and did not indicate directions for future research, which is required in classical scientific publications.
Round 2
Reviewer 2 Report
Dear Authors
The introduction is more straightforward in the current version, but methodological issues remain in the methods and results sections. Researchers need to define their research objectives without getting too bogged down in minor details that can confuse readers. Additionally, more clear explanations are needed in the methods and results chapters.
Please clarify which aim you are referring to. Is it the development of the Real Estate Market Sentiment Index or the Empirical Research Model?
It is critical to provide a detailed breakdown of the research design. This includes how the different components such as terms, objectives, methods, and results are connected. This will ensure clarity and accuracy in the research findings.
The first heading in the literature review section should be “Short-Term Control Policy”
It is advisable to relocate and integrate all rephrased definitions and terminology from the Methods section into the literature review chapter. Separating search purposes needs to be clarified. For research to be effective, it is essential to clearly and specifically state the purpose of the study in the title, abstract, and introduction, as outlined in the methods chapter. Can you ensure that the purpose of your research is explicitly stated in these sections?
In scientific research writing, the goal typically follows the study's problem, gaps, and justifications but comes before the research contribution. It should not be placed after the literature review. Then, the goals that achieve this goal can be detailed.
Separating hypotheses could be more apparent.
The introduction has already provided a clear explanation of the problem, so there is no need to repeat it here. It would be beneficial to combine the ideas into two paragraphs within the introduction. The hypotheses or assumptions can be stated directly in the methods chapter.Such a potentially unrealistic assumption arises from the fact that __
Based on these assumptions, hypotheses were developed: __
Please review multiple paragraphs for any signs of bias, such as:
It is precisely due to these characteristic hallmarks that this article positions China as an exemplary research arche-type, aspiring to furnish invaluable experiential insights to other nations contemplating the adoption or refinement of their own purchase restriction policies. Remove “invaluable”
This paper applies a unique state-space model to analyze consumer sentiment in the real estate market. Remove “unique”
Reviewer 3 Report
Accept
Round 3
Reviewer 2 Report
Dear authors
Thank you.